# Replacement Learning: Training Vision Tasks with Fewer Learnable Parameters

## Abstract

Traditional end-to-end deep learning models often enhance feature representation and overall performance by increasing the depth and complexity of the network during training. However, this approach inevitably introduces issues of parameter redundancy and resource inefficiency, especially in deeper networks. While existing works attempt to skip certain redundant layers to alleviate these problems, challenges related to poor performance, computational complexity, and inefficient memory usage remain. To address these issues, we propose an innovative training approach called Replacement Learning, which mitigates these limitations by completely replacing all the parameters of the frozen layers with only two learnable parameters. Specifically, Replacement Learning selectively freezes the parameters of certain layers, and the frozen layers utilize parameters from adjacent layers, updating them through a parameter integration mechanism controlled by two learnable parameters. This method leverages information from surrounding structures, reduces computation, conserves GPU memory, and maintains a balance between historical context and new inputs, ultimately enhancing overall model performance. We conducted experiments across four benchmark datasets, including CIFAR-10, STL-10, SVHN, and ImageNet, utilizing various architectures such as CNNs and ViTs to validate the effectiveness of Replacement Learning. Experimental results demonstrate that our approach reduces the number of parameters, training time, and memory consumption while completely surpassing the performance of end-to-end training.

## 1 Introduction

Updating learnable parameters is a core component of training deep learning models Yang et al. (2019). Currently, the primary mechanism for updating parameters in these frameworks is global backpropagation Mostafa et al. (2018), a technique widely applied in various fields, including computer vision Yoo (2015); Voulodimos et al. (2018), natural language processing Goldberg (2016; 2017), and speech processing Ahmad et al. (2004); Chauvin & Rumelhart (2013). However, the increase in network depth and model complexity during training leads to a rapid expansion in the computation time and parameter demands required by global backpropagation Nawi et al. (2008). This rise in computational and memory costs inevitably poses significant challenges to GPU processing capabilities and memory capacity Bragagnolo et al. (2022). Furthermore, the high similarity in learning patterns between adjacent layers in deep learning models Kleinman et al. (2021) results in parameter redundancy and inefficient resource usage throughout the computation process. With the growing popularity of large models, there is an urgent need to find training methods that can shorten computation time, reduce GPU memory usage, and still ensure model performance.

To address these challenges, researchers have explored alternatives to traditional backpropagation, such as feedback alignment Lillicrap et al. (2014); Nøkland (2016), forward gradient learning Dellaferrera & Kreiman (2022); Ren et al. (2022), and local learning Su et al. (2024a;b). These methods aim to reduce computational overhead by updating weights without relying entirely on backpropagation. However, these improvements do not completely address the inherent short-sightedness: the separation into blocks can make each part of the network only focus on its local objectives, possibly ignoring the overall objectives of the network. This can lead to the discarding of globally beneficial information due to the lack of inter-block communication. Additionally, self-attention layers in Vision Transformers (ViT) Dosovitskiy et al. (2021) exhibit high correlations between adjacent

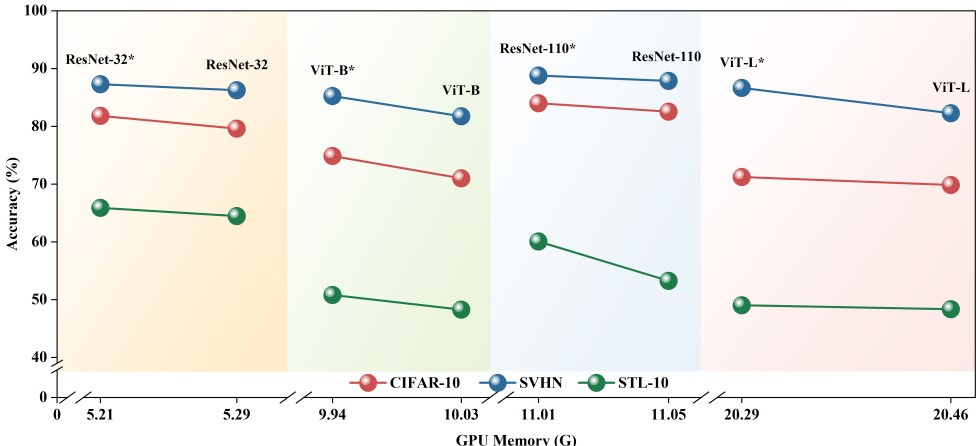

Figure 1: Comparison between different backbones with the training of Replacement Learning and end-to-end training regarding GPU Memory and Accuracy. Results are obtained using ViT-B, ViT-L, ResNet-32, and ResNet-110 on CIFAR-10, SVHN, and STL-10.

layers, leading to the development of the skip attention method Venkataramanan et al. (2023). This method reuses attention calculations to reduce computation but risks propagating errors, potentially degrading performance, and causing overfitting. Thus, both alternative backpropagation techniques and skip attention strategies struggle to maintain model performance while improving efficiency.

In this paper, we introduce a novel learning approach called Replacement Learning, designed to address the challenge of maintaining model performance while reducing computational overhead and resource consumption. Replacement Learning freezes specific layers, and during backpropagation, these frozen layers utilize parameters from adjacent layers, updating them through a parameter integration mechanism controlled by two learnable parameters, further optimizing efficiency. Considering that parameters from adjacent layers, if solely derived from either shallow or deep layers, often fail to simultaneously enable frozen layers to excel in learning both local and global features, the frozen layers are designed to leverage parameters from both preceding and succeeding layers, which facilitates a more effective fusion of low-level and high-level information. Moreover, to prevent the continuity of feature extraction from being disrupted, we introduce optimized interval settings for frozen layers in Replacement Learning, striking an effective balance between computational efficiency and performance. Replacement Learning significantly reduces the number of parameters while allowing frozen layers to incorporate information from adjacent layers. Balancing historical context with new inputs through the two learnable parameters, Replacement Learning enhances the model's overall performance. The effectiveness of Replacement Learning has been rigorously validated on multiple benchmark image classification datasets, including CIFAR-10 Krizhevsky et al. (2009), STL-10 Coates et al. (2011), SVHN Netzer et al. (2011), and ImageNet Deng et al. (2009), across various architectures such as CNNs and ViTs. Experimental results demonstrate that Replacement Learning not only reduces the number of parameters, training time, and memory usage but also outperforms end-to-end Rumelhart et al. (1985) training approaches.

We summarize our contributions as follows:

- We propose a novel learning approach, Replacement Learning, which reduces the number of parameters, training time, and memory consumption while obtaining better performance than end-to-end training Rumelhart et al. (1985).

- Replacement Learning has strong versatility, and as a universal training method, it can be seamlessly applied across architectures of varying depths and performs robustly on diverse datasets.

- The effectiveness of Replacement Learning has been validated on commonly used datasets such as CIFAR-10 Krizhevsky et al. (2009), STL-10 Coates et al. (2011), SVHN Netzer et al. (2011), and ImageNet Deng et al. (2009) on both CNNs and ViTs structures, and its performance has fully surpassed that of end-to-end training Rumelhart et al. (1985).

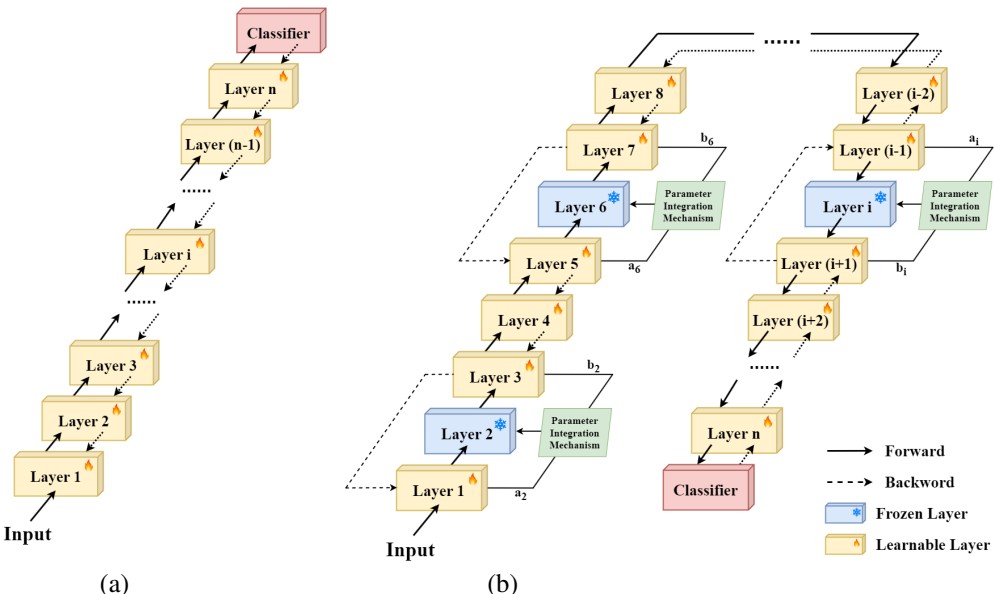

Figure 2: Comparison of (a) end-to-end backpropagation and (b) our proposed Replacement Learning.

## 2 RELATED WORK

### 2.1 ALTERNATIVES TO BACKPROPAGATION

To address the limitations of backpropagation, such as high computational cost, various alternative methods have been proposed, including target propagation Lee et al. (2015); Bartunov et al. (2018), feedback alignment Lillicrap et al. (2014); Nøkland (2016), and decoupled neural interfaces (DNI) Jaderberg et al. (2017). These approaches bypass traditional global backpropagation by directly propagating errors to individual layers, reducing memory usage and enhancing efficiency. Forward gradient learning Dellaferrera & Kreiman (2022); Ren et al. (2022) offers a new paradigm for training deep networks more effectively. Local learning Zhang et al. (2024); Zhu et al. (2024) segments the network into smaller, independently trained modules, optimizing local objectives to lower computational demands while preserving some global features Su et al. (2024a;b). However, excessive segmentation can lead to coordination issues, harming overall performance, especially on complex datasets like ImageNet.

### 2.2 UTILIZING SURROUNDING LAYERS

Leveraging the high similarity in learning conditions of surrounding layers, researchers have solved many problems in deep learning. Some studies have applied Residual Networks (ResNets) He et al. (2016), by adding a shortcut connection to the activation function of the next layer, this identity mapping enables ResNet to address the issues of degradation Philipp et al. (2018); Borawar & Kaur (2023), enhancing both the convergence speed and accuracy of the network Zhang et al. (2019); Allen-Zhu & Li (2019). Additionally, some researchers have proposed skipping attention, reusing the self-attention calculations from one layer in the approximations for attention in subsequent layers, achieving higher throughput Venkataramanan et al. (2023). However, due to the repeated use of prior layers, this method carries the risk of error propagation and could potentially cause losses during the learning process, impacting the model's generalization ability.

## 3 METHOD

### 3.1 PREPARATIONS

To begin, we briefly introduce the forward and backward propagation processes of the traditional end-to-end training model Rumelhart et al. (1985) to clarify the background. Let us assume that the depth of a network is $n$. For an input image $x$, the forward propagation process through $n$ layers of the neural network is as follows:

$$h_0 = x \tag{1}$$

$$h_i = f_i(h_{i-1}; \theta_i), i = 1, 2, \cdots, n \tag{2}$$

here, $h_i$ represents the activation value of the $i$-th layer, $f_i$ is the forward computation function of the $l$-th layer, and $\theta_i$ are the learnable parameters of the $i$-th layer.

Once the entire forward propagation process of the network is completed, we can calculate the loss $\mathcal{L}$ based on the label $y$:

$$\mathcal{L} = \mathcal{L}(h_n, y) \tag{3}$$

After the loss $\mathcal{L}$ is calculated, the network can perform backward propagation to update the parameters for each layer. The gradient computation and parameter updates for each layer are as follows:

$$\delta_n = \frac{\partial \mathcal{L}}{\partial h_n} \tag{4}$$

$$\delta_i = \delta_{i+1} \times \frac{\partial h_{i+1}}{\partial h_i}, i = n - 1, n - 2, \cdots, 1 \tag{5}$$

$$\frac{\partial \mathcal{L}}{\partial \theta_i} = \delta_i \times \frac{\partial h_i}{\partial \theta_i} \tag{6}$$

$$\delta_i = \delta_i - \eta \times \frac{\partial \mathcal{L}}{\partial \theta_i} \tag{7}$$

where $\eta$ denotes the learning rate of the network, $\delta_i$ and $\delta_n$ are the gradients of the i-th and n-th layers, respectively.

### 3.2 REPLACEMENT LEARNING

Traditional end-to-end training Rumelhart et al. (1985) is the mainstream method for training models. However, as the network depth and model complexity increase during training, all layers are involved in the optimization process. Combined with the high computational complexity of chain rule-based gradient calculations in both forward and backward propagation, this results in a large number of parameters and high demands on computation time and resources. Furthermore, given the high similarity of features between adjacent layers, it becomes unnecessary for every layer to participate in parameter updates during backpropagation Rumelhart et al. (1985), which results in parameter redundancy and inefficient training.

To address these issues, we propose Replacement Learning. The innovation of Replacement Learning lies in the mechanism of periodically freezing a layer's parameters, denoted as $\theta_i$, utilizing parameters from adjacent layers, and updating them through a parameter integration mechanism controlled by two learnable parameters. This idea is inspired by Exponential Moving Average (EMA) He et al. (2020), where two learnable parameters are introduced to balance the historical context with new inputs.

When the adjacent layers used come from the preceding layers of $\theta_i$, they tend to perform well in learning local features due to capturing preceding contextual information. Still, they are less effective in acquiring global high-level semantic information. Conversely, when the adjacent layers are from the succeeding layers, the deeper layer parameters can ensure the learning of higher-level semantic and global features but perform poorly in extracting fine-grained features and capturing low-level details. Therefore, we opt to simultaneously utilize both preceding and succeeding layers, $\theta_{i-1}$ and $\theta_{i+1}$, and integrate their parameters, which facilitates a better fusion of low-level and high-level information, thereby enhancing the overall performance of the model.

We incorporate learnable parameters for $\theta_{i-1}$ and $\theta_{i+1}$, $a_i$ and $b_i$. During the forward propagation, the parameters of $\theta_i$ are replaced by parameter integration results based on the parameters of $\theta_{i-1}$

and $\theta_{i+1}$. Among them, $a_i$ and $b_i$ play a role in dynamically adjusting the contributions of $\theta_{i-1}$ and $\theta_{i+1}$. In the backpropagation process, $\theta_i$ does not participate in the parameter updates from gradient descent. We will explain this process again using the forward and backward propagation steps of our method, and the specific implementation process refers to Algorithm 1 in the Appendix. In the network, for every $k$ layer there is a frozen layer, It should be noted that if the $n$-th layer is the final layer and $n$ is an integer multiple of $k$, this layer will not be frozen. The set of indices for the frozen layers is:

$$\mathcal{F} = \{\, i \mid i \bmod k = 0 \,\}, \quad i = k, 2k, 3k, \ldots \tag{8}$$

During the forward propagation process, the propagation through non-frozen layers follows the same process as described in Eq.2, while the parameters and computation process for the frozen layers are as follows:

$$\theta_i = a_i \times \theta_{i-1} + b_i \times \theta_{i+1} \tag{9}$$

$$h_i = f_i(h_{i-1}; \theta_i) \tag{10}$$

where, $h_i$ represents the activation value of the $i$-th layer, $f_i$ is the forward computation function of the $i$-th layer, and $\theta_i$ are the learnable parameters of the $i$-th layer.

After completing the forward propagation through all the layers of the network, we can calculate the loss $\mathcal{L}$ as described in Eq.3. Subsequently, layer-by-layer backward propagation can begin. The backward propagation process for non-frozen layers is consistent with what is described in Eq.5, Eq.6, and Eq.7. The backward propagation process for the frozen layers is as follows:

First, we calculate the error term for it:

$$\delta_i = \delta_{i+1} \times \frac{\partial h_{i+1}}{\partial h_i} \tag{11}$$

Subsequently, we compute the gradients for $a_i$ and $b_i$ as follows:

$$\frac{\partial \mathcal{L}}{\partial a_i} = \delta_i \times \frac{\partial h_i}{\partial \theta_i} \times \theta_{i-1}, \frac{\partial \mathcal{L}}{\partial b_i} = \delta_i \times \frac{\partial h_i}{\partial \theta_i} \times \theta_{i+1} \tag{12}$$

After the gradient calculations are complete, we update the parameters $a_i$ and $b_i$:

$$a_i = a_i - \eta \times \frac{\partial \mathcal{L}}{\partial a_i}, b_i = b_i - \eta \times \frac{\partial \mathcal{L}}{\partial b_i} \tag{13}$$

Finally, the error propagates to the adjacent layers:

$$\delta_{i-1} = \delta_i \times \left( \frac{\partial h_i}{\partial h_{i-1}} + \frac{\partial h_i}{\partial \theta_i} \times a_i \times \frac{\partial \theta_i}{\partial \theta_{i-1}} \right) \tag{14}$$

$$\delta_{i+1} = \delta_{i+1} + \delta_i \times \frac{\partial h_i}{\partial \theta_i} \times b_i \times \frac{\partial \theta_i}{\partial \theta_{i+1}} \tag{15}$$

Essentially, the proposed Replacement Learning replaces the complete set of parameters in certain layers with only two learnable parameters, effectively mitigating the issues of high computational cost, long training time, and GPU memory consumption inherent in traditional end-to-end training Rumelhart et al. (1985). Moreover, by employing the parameter integration mechanism, Replacement Learning enhances the network's overall performance. Furthermore, Replacement Learning demonstrates strong versatility, as it can be seamlessly applied across architectures of varying depths and performs robustly on diverse datasets. This versatility is crucial for efficiently training deeper and more complex deep learning models.

## 4 EXPERIMENTS

### 4.1 EXPERIMENTAL SETUP

We conduct experiments using four widely adopted datasets: CIFAR-10 Krizhevsky et al. (2009), SVHN Netzer et al. (2011), STL-10 Coates et al. (2011), and ImageNet Deng et al. (2009), with

Table 1: Perfomance of different backbones. The * means the usage of our Replacement Learning.

| Datasets | Backbone | Test Accuracy | GPU Memory | Time (Each epoch) |
|----------|----------|---------------|------------|-------------------|
| CIFAR-10 | ViT-B | 72.23 | 10.03G | 7.37s |
| | **ViT-B\*** | **72.86 (↑ 0.63)** | **9.94G (↓ 0.90%)** | **7.36s (↓ 0.14%)** |
| | ViT-L | 69.85 | 20.46G | 16.68s |
| | **ViT-L\*** | **71.23 (↑ 1.38)** | **20.29G (↓ 0.83%)** | **16.58s (↓ 0.60%)** |
| | ResNet-32 | 79.60 | 5.29G | 3.63s |
| | **ResNet-32\*** | **81.82 (↑ 2.22)** | **5.21G (↓ 1.51%)** | **3.46s (↓ 4.68%)** |
| | ResNet-110 | 83.53 | 11.05G | 10.68s |
| | **ResNet-110\*** | **83.99 (↑ 0.46)** | **11.01G (↓ 0.36%)** | **9.62s (↓ 9.93%)** |
| SVHN | ViT-B | 81.74 | 10.03G | 11.02s |
| | **ViT-B\*** | **85.15 (↑ 3.41)** | **9.94G (↓ 0.90%)** | **10.96s (↓ 0.54%)** |
| | ViT-L | 82.27 | 20.46G | 25.41s |
| | **ViT-L\*** | **84.77 (↑ 2.50)** | **20.29G (↓ 0.83%)** | **25.19s (↓ 0.87%)** |
| | ResNet-32 | 86.24 | 5.29G | 5.63s |
| | **ResNet-32\*** | **87.30 (↑ 1.06)** | **5.21G (↓ 1.51%)** | **5.56s (↓ 1.24%)** |
| | ResNet-110 | 87.86 | 11.05G | 15.96s |
| | **ResNet-110\*** | **88.18 (↑ 0.32)** | **11.01G (↓ 0.36%)** | **14.80s (↓ 11.78%)** |
| STL-10 | ViT-B | 48.27 | 10.03G | 1.50s |
| | **ViT-B\*** | **50.81 (↑ 2.54)** | **9.94G (↓ 0.90%)** | **1.49s (↓ 0.67%)** |
| | ViT-L | 48.35 | 20.46G | 2.94s |
| | **ViT-L\*** | **49.03 (↑ 0.68)** | **20.29G (↓ 0.83%)** | **2.90s (↓ 1.36%)** |
| | ResNet-32 | 64.47 | 5.29G | 1.09s |
| | **ResNet-32\*** | **64.49 (↑ 0.02)** | **5.21G (↓ 1.51%)** | **1.06s (↓ 2.75%)** |
| | ResNet-110 | 53.25 | 11.05G | 2.24s |
| | **ResNet-110\*** | **60.08 (↑ 6.83)** | **11.01G (↓ 0.36%)** | **1.81s (↓ 19.20%)** |

Vision Transformer (ViT) Dosovitskiy et al. (2021) and ResNets He et al. (2016) of varying depths, serve as the network architectures.

During the experiment, we do not utilize pre-trained models. Instead, we train from scratch. We set $k = 4$ as the interval for the parameter integration mechanism. Apart from the frozen layers, all other layers compute the loss using gradient descent and update the parameters via backpropagation Rumelhart et al. (1985).

### 4.2 COMPARISON WITH THE SOTA RESULTS

#### 4.2.1 RESULTS ON SMALL IMAGE CLASSIFICATION BENCHMARKS

We start by assessing the accuracy performance of our method using the CIFAR-10 Krizhevsky et al. (2009), SVHN Netzer et al. (2011), and STL-10 Coates et al. (2011) datasets. As illustrated in Table 1, the performance of Replacement Learning significantly exceeds the performance trained using end-to-end Rumelhart et al. (1985) training on all structures.

Replacement Learning, on the CIFAR-10 dataset, considerably improves Test Accuracy across various backbones. In the network of ViT-B and ViT-L Dosovitskiy et al. (2021), we record an improvement in Test Accuracy, from 72.23, 69.85 to 72.86, 71.23. In the relatively shallower network of ResNet-32 He et al. (2016), the Test Accuracy rises from 79.60 to 81.82. Even though the performance in the comparatively deeper network, ResNet-110 He et al. (2016), is somewhat inferior due to the inherent need for more global information in such networks, our method still delivers exceptional performance. It achieves approximately a 0.46 improvement, underscoring the robust effectiveness of Replacement Learning in deeper networks.

When applied to other datasets, Replacement Learning can also increase Test Accuracy by at least 3.41, 2.50, 1.06, and 0.32 on the STL-10 dataset Coates et al. (2011). On the SVHN Netzer et al. (2011) dataset, our improvements over the four backbones also surpass 2.54, 0.68, 0.02, and 6.83,

Table 2: Results on the validation set of ImageNet

| Backbone | Top1 Accuracy | Top5 Accuracy | GPU Memory | Time (Each epoch) |
|---|---|---|---|---|
| ViT-B | 49.88 | 73.86 | 19.03G | 4673s |
| **ViT-B*** | **50.82 (↑ 0.94)** | **74.73 (↑ 0.87)** | **18.63G (↓ 2.10%)** | **4578s (↓ 2.03%)** |
| ResNet-34 | 55.49 | 79.82 | 17.46G | 3493s |
| **ResNet-34*** | **57.06 (↑ 1.57)** | **80.77 (↑ 0.95)** | **17.03G (↓ 2.46%)** | **3391s (↓ 2.92%)** |
| ResNet-101 | 53.10 | 77.75 | 17.03G | 10268s |
| **ResNet-101*** | **54.76 (↑ 1.66)** | **78.74 (↑ 0.99)** | **16.38G (↓ 3.82%)** | **10029s (↓ 2.33%)** |
| ResNet-152 | 51.49 | 75.87 | 23.17G | 14478s |
| **ResNet-152*** | **53.18 (↑ 1.69)** | **76.91 (↑ 1.04)** | **22.90G (↓ 1.17% )** | **14159s (↓ 2.21%)** |

respectively. As can be seen, the improvement in our Replacement Learning substitution to all backbones is quite remarkable and comparable.

Other significant advantages of Replacement Learning can also be seen in Table 1, integrating Replacement Learning into various backbone architectures demonstrates a consistent reduction in GPU memory usage and training time across multiple datasets while maintaining or improving model performance. On the CIFAR-10 Krizhevsky et al. (2009), Replacement Learning integration leads to notable reductions in GPU memory consumption and training time. Specifically, GPU memory usage is reduced by 0.90% for ViT-B Dosovitskiy et al. (2021) and 0.83% for ViT-L Dosovitskiy et al. (2021) models, while ResNet-32 and ResNet-110 He et al. (2016) see reductions of 1.51% and 0.36%, respectively. Training time per epoch is also decreased, with ViT-B and ViT-L Dosovitskiy et al. (2021) showing reductions of 0.14% and 0.60%, respectively, and ResNet-32 and ResNet-110 He et al. (2016) benefiting from reductions of 4.68% and 9.93% per epoch. Similar trends are observed on the SVHN Netzer et al. (2011) and STL-10 Coates et al. (2011), where Replacement Learning consistently reduces GPU memory usage and training time across various backbones, reinforcing its effectiveness in optimizing computational efficiency.

### 4.2.2 RESULTS ON IMAGENET

We further validate the effectiveness of Replacement Learning on ImageNet Deng et al. (2009) using four backbones of ViT-B Dosovitskiy et al. (2021), ResNet-34, ResNet-101, and ResNet-152 He et al. (2016) as depicted in Table 2. When we employ ViT-B Dosovitskiy et al. (2021) as the backbone, it achieves merely a Top1 Accuracy of 49.88 and a Top5 Accuracy of 73.86. However, with the usage of our Replacement Learning, the Top1 Accuracy increases by 0.94, and the Top5 Accuracy rises by 0.87. As illustrated in Table 2, the performance is below when we use ResNet-34, ResNet-101, and ResNet-152 He et al. (2016) as backbones. After using our Replacement Learning, the Top1 Accuracy of these three backbone networks can be increased by 1.57, 1.66, and 1.69, Top5 Accuracy can be increased by 0.95, 0.99, and 1.04, respectively, compared to the original. Not only that, for GPU and training time, Replacement Learning has varying degrees of memory savings on all four models and can save an average training time of 2%-3% each epoch. These results underscore the effectiveness of our Replacement Learning on the large-scale ImageNet dataset, even when using deeper networks.

### 4.3 ABLATION STUDY

### 4.3.1 COMPARISON OF FEATURES IN DIFFERENT UPDATING LAYERS

To showcase the advanced capabilities of Replacement Learning, we conduct feature map Selvaraju et al. (2017) analyses with ResNet-32 He et al. (2016) on different configurations, including end-to-end training Rumelhart et al. (1985), training with parameters updated by the preceding layer, training with parameters updated by the succeeding layer, and our Replacement Learning. The resulting figures detailing these feature maps can be found in Figure 4.3.1. Upon analyzing them, we can observe that (a) is concentrated in specific regions, indicating the presence of significant information within those areas. Conversely, after the fusion of (b) and (c), (d) captures more com-

prehensive global features, including localized edge features. It follows that the outstanding ability of Replacement Learning to capture global features.

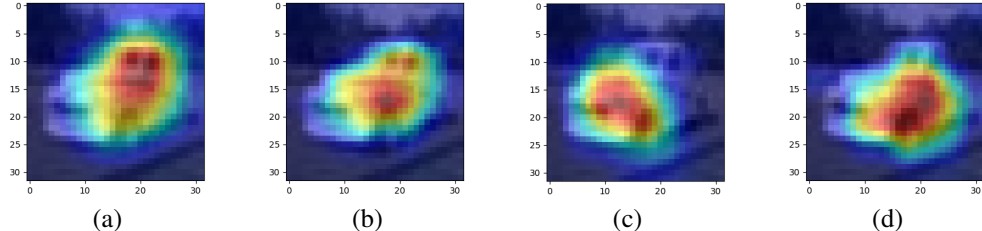

|     (a)     |     (b)     |     (c)     |     (d)     |

Figure 3: Visualization of feature maps. (a) Feature map of end-to-end training. (b) Feature map of training with parameters updated by the preceding layer. (c) Feature map of training with parameters updated by the succeeding layer. (d) Feature map of Replacement Learning, parameters updated by both the preceding and succeeding layers.

### 4.3.2    REPRESENTATION SIMILARITY ANALYSIS

To further demonstrate Replacement Learning's effectiveness, we conduct Centered Kernel Alignment (CKA) Kornblith et al. (2019) experiments. On the CIFAR-10 Krizhevsky et al. (2009), we compute the similarity between layers for both the end-to-end training Rumelhart et al. (1985) and training of Replacement Learning, with ResNet-32 He et al. (2016) serving as a representative case. From Figure 4.3.2, we observe that the feature similarity across layers in (b) is generally lower, except between the frozen layers (k=4, indicating that every 4th layer exhibits high feature similarity). Experimental results highlight three key advantages of Replacement Learning. First, in (a), the re-

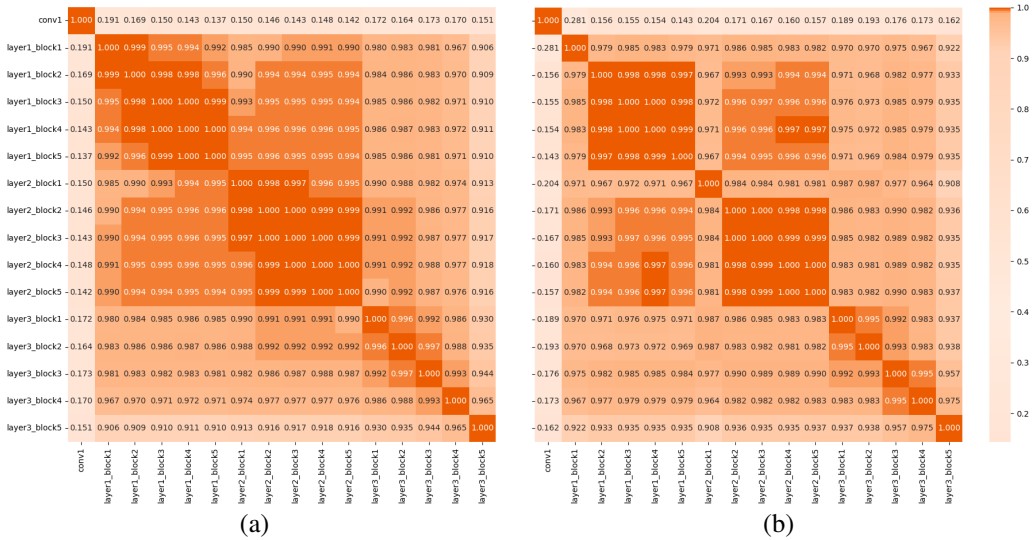

Figure 4: Visualization of similarity matrixes. (a) Similarity matrix of end-to-end training. (b) Similarity matrix of Replacement Learning.

sult of end-to-end training Rumelhart et al. (1985) shows a gradual, smooth decrease in inter-layer similarity, suggesting progressively abstract features with depth. In contrast, (b) reveals more significant fluctuations, indicating that Replacement Learning captures more diverse features at different depths. Second, higher similarity in (a) suggests potential feature redundancy, limiting performance, while (b)'s lower similarity implies more distinct feature extraction, beneficial for complex tasks. Lastly, (a) may overfit the training set due to concentrated features, while (b)'s lower similarity, especially in deeper layers, enhances generalization.

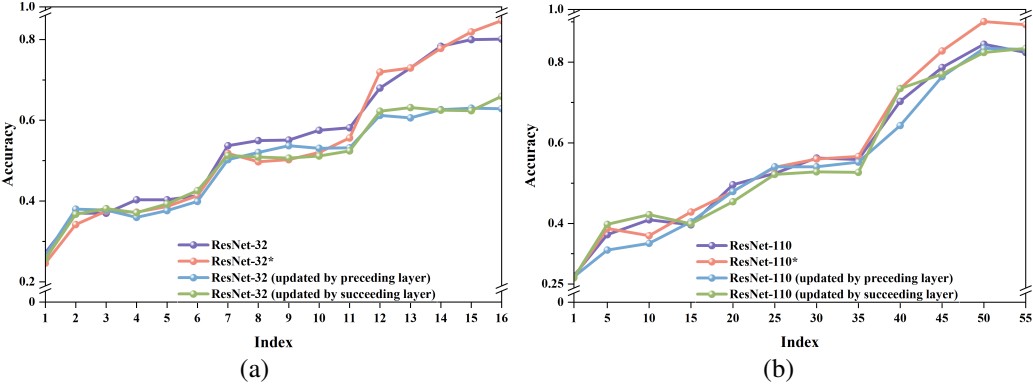

Figure 5: Comparison of layer-wise linear separability. (a) Linear Separability of RestNet-32 on CIFAR-10. (b) Linear Separability of RestNet-110 on CIFAR-10.

### 4.3.3 DECOUPLED LAYER ACCURACY ANALYSIS

We have demonstrated that Replacement Learning achieves accuracy comparable to the end-to-end training Rumelhart et al. (1985). To further analyze Replacement Learning's impact, we train a linear classifier for each layer. Results are shown in Figure 4.3.3 using ResNet-32 and ResNet-110 He et al. (2016) as the baselines. The outcomes suggest that the selective freezing of layers combined with the parameter integration mechanism updates effectively enhances the robustness and generalization capability of the model. Specifically, Replacement Learning demonstrates higher accuracy in both ResNet-32 and ResNet-110 He et al. (2016), with the advantage being more pronounced in the deeper ResNet-110 network. These findings validate the effectiveness of the proposed Replacement Learning in deep neural networks, particularly in managing the interactions between layers. Overall, the experimental results support our hypothesis that improving inter-layer information transmission mechanisms can significantly enhance the performance of deep neural networks without increasing model complexity.

## 4.4 PERFORMANCE STUDY

### 4.4.1 PARAMETER ANALYSIS

To investigate the factors contributing to the learnable parameter reduction in Replacement Learning, we consider a network model with $L$ layers, where the number of learnable parameters in the $i$-th layer is denoted as $P_i$. In end-to-end training Rumelhart et al. (1985), all layers' parameters are simultaneously optimized, resulting in a total parameter count of $P = \sum_{i=1}^{L} P_i$. In contrast, with Replacement Learning, the parameters in frozen layers are not independently updated. Instead, they are adjusted using two learnable parameters that approximate the impact of adjacent layers. Thus, the total parameter count under Replacement Learning becomes $P' = P - \sum_{i=1}^{L} P_{i+nk} + 2, \ n \geq 0$. Although two additional learnable parameters are introduced, the total number of learnable parameters is reduced by $\sum_{i=1}^{L} P_{i+nk} - 2$ compared to end-to-end training, thereby decreasing the computational demand for parameter updates.

To further analyze the range and patterns of learnable parameter reduction, let the number of parameters in all layers $P_i$ have a maximum value of $P_{\max}$ and a minimum value of $P_{\min}$ across all layers, $P_{\min} \leq P_i \leq P_{\max}, \quad \forall i = 1, 2, \ldots, L$. In the case where all activated layers have the minimum number of parameters, the reduction in parameters is given by $\sum_{i=1}^{L} P_{i+nk} - 2 \geq \frac{L}{k} \times P_{\min} - 2$. While in the case where all activated layers have the maximum number of parameters, the reduction is $\sum_{i=1}^{L} P_{i+nk} - 2 \leq \frac{L}{k} \times P_{\max} - 2$. As the network becomes extremely deep, the ratio $\frac{L}{k}$ increases, making the remaining learnable parameters in the new model significantly lower compared to the original model. The upper and lower limits are:

$$\lim_{L \to \infty} \left( \frac{L}{k} \times P_{\max} - 2 \right) \approx \frac{L}{k} \times P_{\max} - 2 \tag{16}$$

$$\lim_{L \to \infty} \left( \frac{L}{k} \times P_{\min} - 2 \right) \approx \frac{L}{k} \times P_{\min} - 2 \tag{17}$$

Therefore, when all activated layers have the maximum number of parameters, the reduction is given by: $\sum_{i=1}^{L} P_{i+nk} - 2 \leq \frac{L}{k} \times P_{\max} - 2$, and when all activated layers have the minimum number of parameters, the reduction is $\sum_{i=1}^{L} P_{i+nk} - 2 \geq \frac{L}{k} \times P_{\min} - 2$. As $L$ approaches infinity, the upper and lower bounds converge to a multiple of the parameter values in the activated layers, scaled by $\frac{L}{k}$.

### 4.4.2 COMPLEXITY ANALYSIS

Tables 1 and 2 compare the GPU memory consumption of different network architectures at varying depths using end-to-end training Rumelhart et al. (1985) and Replacement Learning. To explain why Replacement Learning uses less GPU memory, we analyze the computational complexity of the two methods. In E2E training, all parameters participate in forward propagation, resulting in a complexity of $L \times O(F)$. Backward propagation, requiring gradient calculations, has approximately twice this complexity, making the total $3 \times L \times O(F)$. In Replacement Learning, forward propagation also has a complexity of $L \times O(F)$. During backward propagation, only $L - \frac{L-1}{k}$ unfrozen layers are optimized, each with a backward complexity of $2 \times O(F)$, leading to $(L - \frac{L-1}{k}) \times 2 \times O(F)$. The frozen layers involve only two learnable parameters, with a negligible backward complexity of $2 \times \frac{L-1}{k} \times O(1)$. Thus, the total computational complexity is $L \times O(F) + (L - \frac{L-1}{k}) \times 2 \times O(F) + 2 \times \frac{L-1}{k} \times O(1) \approx (3L - 2 \times \frac{L-1}{k}) \times O(F)$.

Compared to end-to-end training, the complexity of Replacement Learning is reduced by $2 \times \frac{L-1}{k} \times O(F)$. Based on the characteristics of deep learning, we analyze the upper and lower bounds, as well as the limit, of the complexity reduction $2 \times \frac{L-1}{k} \times O(F)$. The upper bound is achieved when $k = 1$, meaning no layers are frozen. In this case, the complexity reduction is:

$$2 \times \frac{L-1}{k} \times O(F) \bigg|_{k=1} = (2L - 2) \times O(F) \tag{18}$$

While the lower bound is obtained when $k = L - 1$, with only one active layer. Here, the complexity reduction is:

$$2 \times \frac{L-1}{k} \times O(F) \bigg|_{k=L-1} = 2 \times O(F) \tag{19}$$

As $L \to \infty$, the limit of the complexity reduction depends on $k$. If $k$ is a constant, the complexity reduction increases linearly with $L$. If $k \approx L$, the reduction converges to $O(F)$, indicating a stable reduction ratio.

## 5 CONCLUSION

This paper introduces a novel learning approach called Replacement Learning to address the problem of maintaining model performance while reducing computational overhead and resource consumption. Replacement Learning effectively reduces the parameter count while enabling frozen layers to integrate information from neighboring layers. Utilizing two learnable parameters to balance historical context and new inputs boosts the model's overall performance. We apply Replacement Learning to various model structures and evaluate its performance on four widely used datasets across different deep network structures. The results demonstrate that our proposed Replacement Learning not only reduces training time and GPU usage but also consistently outperforms end-to-end training in terms of overall performance.

**Limitations and future work:** Although our proposed Replacement Learning can reduce the number of parameters to be computed, save memory, and decrease training time while outperforming end-to-end training, it has only been applied to image-based tasks and has not yet been extended to other large models, such as those in natural language processing or multimodal settings. In future work, we plan to explore the impact of Replacement Learning on other tasks to achieve a more comprehensive evaluation of the model's effectiveness.

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

# A  APPENDIX

## A.1  EXPERIMENT IMPLEMENT DETAILS

In our experiments on CIFAR-10 Krizhevsky et al. (2009), SVHN Netzer et al. (2011), and STL-10 Coates et al. (2011) datasets, we utilize the AdamW optimizer Loshchilov & Hutter (2017) with a weight decay factor of 1e-4 for ViT-B, ViT-L Dosovitskiy et al. (2021), ResNet-32, and ResNet-110 He et al. (2016). We employ batch sizes of 1024 for CIFAR-10 Krizhevsky et al. (2009), SVHN Netzer et al. (2011), and STL-10 Coates et al. (2011). The training duration spans 250 epochs, starting with initial learning rates of 0.01, following a cosine annealing scheduler Loshchilov & Hutter (2016).

For ImageNet Deng et al. (2009), We use the AdamW optimizer Loshchilov & Hutter (2017) with a weight decay factor of 1e-4. Different hyperparameters are used for each architecture: batch size is 128 for ViT-B Dosovitskiy et al. (2021) and ResNet-34 He et al. (2016), and batch size is 32 for ResNet-101 and ResNet-152 He et al. (2016). Training lasts 100 epochs with initial learning rates of 0.04 for ViT-B Dosovitskiy et al. (2021) and ResNet-34 He et al. (2016), and 0.01 for ResNet-101 and ResNet-152 He et al. (2016).

We recognize that in the Transformer Encoder of the ViT Dosovitskiy et al. (2021) architecture, one layer consists of an MLP and a Multi-Head Attention. When freezing layers, we freeze only the gradients of the Multi-Head Attention, without altering the gradient descent of the MLP during forward propagation. For the ResNet He et al. (2016) architecture, we refer to each residual block as a layer, where each layer is composed of two convolutions. The entire layer is frozen during gradient freezing, with the parameters derived from the parameter integration mechanism entering the next layer via the residual connection.

## A.2  GENERALIZATION STUDY

In this section, we aim to investigate the generalization performance of our proposed Replacement Learning. To evaluate its effectiveness, we utilize the checkpoints trained on the CIFAR-10

Table 3: Generalization study. Checkpoints are trained on the CIFAR-10 and tested on the STL-10. The data in the table represents the test accuracy.

| Backbone | Test Accuracy | Backbone | Test Accuracy |
|---|---|---|---|
| ResNet-32 | 36.88 | ViT-B | 28.31 |
| **ResNet-32*** | **37.95 (↑ 1.07)** | **ViT-B*** | **30.14 (↑ 1.83)** |
| ResNet-110 | 39.19 | ViT-L | 26.25 |
| **ResNet-110*** | **39.76 (↑ 0.57)** | **ViT-L*** | **28.02 (↑ 1.77)** |

Krizhevsky et al. (2009) and test them on the STL-10 Coates et al. (2011), taking inspiration from previous work Qu et al. (2021).

As shown in Table 3, with the usage of our Replacement Learning, we witness a significant improvement in test accuracy, surpassing all backbones' end-to-end training Rumelhart et al. (1985). These findings emphasize the efficacy of our Replacement Learning in improving the generalization capabilities of supervised learning, ultimately leading to enhanced overall performance in the image classification task.

### A.3 ALGOTITHM

---

**Algorithm 1** Replace Learning

---

1: Initialize $\theta_l$ for all layers $l = 1$ to $n$
2: Set $k$ as the interval for freezing layers
3: Define frozen layer indices $\mathcal{F} = \{\, l \mid l \bmod k = 0 \,\}$
4: Initialize learnable parameters $a_l$ and $b_l$ for $l \in \mathcal{F}$
5: **for** each mini-batch $(x, y)$ **do**
6:     $h_0 \leftarrow x$
7:     **for** $l = 1$ to $n$ **do**
8:         **if** $l \in \mathcal{F}$ **then**
9:             $\theta_l \leftarrow a_l \times \theta_{l-1} + b_l \times \theta_{l+1}$
10:            $h_l \leftarrow f_l(h_{l-1}; \theta_l)$
11:         **else**
12:            $h_l \leftarrow f_l(h_{l-1}; \theta_l)$
13:         **end if**
14:     **end for**
15:     Compute loss $\mathcal{L} \leftarrow \mathcal{L}(h_n, y)$
16:     Backpropagate to compute gradients
17:     **for** $l = n$ down to 1 **do**
18:         **if** $l \in \mathcal{F}$ **then**
19:            Compute gradients $\frac{\partial \mathcal{L}}{\partial a_l}$ and $\frac{\partial \mathcal{L}}{\partial b_l}$
20:            Update $a_l \leftarrow a_l - \eta \times \frac{\partial \mathcal{L}}{\partial a_l}$
21:            Update $b_l \leftarrow b_l - \eta \times \frac{\partial \mathcal{L}}{\partial b_l}$
22:         **else**
23:            Compute gradient $\frac{\partial \mathcal{L}}{\partial \theta_l}$
24:            Update $\theta_l \leftarrow \theta_l - \eta \times \frac{\partial \mathcal{L}}{\partial \theta_l}$
25:         **end if**
26:     **end for**
27: **end for**

---

