# OpenReview forum: "Replacement Learning: Training Vision Tasks with Fewer Learnable Parameters"
_ICLR.cc/2025/Conference — ICLR 2025 Conference Withdrawn Submission_

### Official Review · Reviewer_S4Bg · 2024-10-23

**Soundness:** 2
**Presentation:** 2
**Contribution:** 3
**Rating:** 5
**Confidence:** 4

**Summary:**

The paper introduced a neural network learning mechanism - Replacement Learning that replaces the parameters of selected layers with just 2 parameters a,b. The output of that layer is computer by a linear model a* activations of previous layer + b* activations of next layer. Given that similar layers in neural network produce correlated outputs, this linear combination approximates the replaced layer's outputs. The method reduces the parameter count, throughput and increases accuracy for image classification datasets(CIFAR-10,STL-10,SVHN and Imagenet-1k).

**Strengths:**

1. Replacement Learning introduces a novel approach for training more efficient models with lesser number of parameters. The parameter integration has good future potential, especially when applied to structured pruning and upon supporting transfer learning.

2. The experiments show good performance for image classification tasks for both convolutional and transformer networks for accuracy, memory and time for 1 epoch.
3. The paper presents ablations to determine the robustness of the method and also presents complexity analysis of the method.
4. The paper addresses the most experimental parameters in the appendix, exhibiting good reproducibility.

**Weaknesses:**

1. The primary concern with the paper is that the experiments are conducted by training networks from scratch. Transfer learning boosts the performance of classification tasks, for instance in the official vit paper[1], VIT B/16 upon transfer learning on CIFAR10 reached an accuracy of 98.13% while the paper achieves only 72.86%, therefore it is important to perform an experiment for examining the effect of replacement learning when transfer learnt. One possible reason why the paper omits trying transfer learning can be initialization of replaced parameters. Parameter Integration $a * \theta_{i-1}+ b* \theta_{i+1}$ is a linear model, this allows approximating a,b values for pre-trained layers in few steps.

2. The paper only shows results on classification, effect of replacement learning on downstream tasks such as object detection and segmentation would strengthen the approach.

3. The flow of gradients is not in a singular direction during replacment learning as suggested by Eq. 15 where $\delta_{i}$ is used for gradient computation of $\delta_{i+1}$. This could cause issues such as vanishing or exploding gradients and must be given a look at.

4.  While the experiments show better time per epoch, the effect of replacement learning on convergence must be studied. This is important as time for 1 epoch can be misleading when the number of epochs to converge is significantly higher than backpropagation.

5. While the paper addresses that only MSA layers were chosen for replacement, where $\theta_{i-1}$ and $\theta_{i+1}$ also MSA layers? This is important as [2] does not guarantee correlation between MSA and MLP layers.

Nitpick: some figure references are wrong, Figures in the ablation display figure numbers 4.3.1, 4.3.2 and 4.3.3 which need to be corrected to figure 3,4,5. Usually it is the placement of caption and label in \begin{figure} that causes this issue.

References:
[1] Dosovitskiy, Alexey. "An image is worth 16x16 words: Transformers for image recognition at scale." arXiv preprint arXiv:2010.11929 (2020).
[2] Venkataramanan, Shashanka, et al. "Skip-attention: Improving vision transformers by paying less attention." arXiv preprint arXiv:2301.02240 (2023).

**Questions:**

I would improve my rating if experiments are performed for points 1 and 4 in weakness and clearing my questions in points, 3 and 5.

I also have a minor question:
 While it is clear both $\theta_{i-1}$ and $\theta_{i+1}$ are useful for approximating $\theta_i$ why not expand it to $\theta_{i-n}$ and $\theta_{i+n}$, this would increase the global context of the computations with minimal increase in parameters?

---

### Official Review · Reviewer_8J38 · 2024-10-30

**Soundness:** 2
**Presentation:** 3
**Contribution:** 3
**Rating:** 5
**Confidence:** 5

**Summary:**

This paper proposes a method of replacement Learning, which reduces computational overhead and resource consumption in deep learning. It enhances model performance while surpassing end-to-end training in efficiency and memory usage. The method has been validated on various datasets and architectures, showing its versatility and potential for broader application.

**Strengths:**

1. The writing of the article is quite good and the article is logical.
2. The experiments on the classification task seem to be  adequate.
3. The charts and graphs are more beautiful and properly presented.
4. Supplementary materials are substantial.

**Weaknesses:**

1. Title: The authors use "Visual Tasks" in the title. "Visual Tasks" include multiple tasks such as classification, detection, segmentation, etc., but it seems that the paper is only validated on the classification task. I suggest adding other tasks to the paper, as has been done in several recent PEFT works [1-3].
[1] 1% vs 100%: Parameter-efficient low rank adapter for dense predictions.
[2] Pro-tuning: Unified prompt tuning for vision tasks.
[3] Adapter is all you need for tuning visual tasks.
[4] Parameter-efficient is not sufficient: Exploring parameter, memory, and time efficient adapter tuning for dense predictions.

2. Related work: the authors perhaps left out some of the most recent work of parameter-efficient fine tuning (PEFT).
3. Experiments: (1) Experiments are performed only on classification tasks; (2) More parameter-efficient fine-tuning methods are available for comparison.

I would consider increasing the score if the authors could provide more convincing comparative experiments.

**Questions:**

See above

---

### Official Review · Reviewer_aFiP · 2024-11-02

**Soundness:** 1
**Presentation:** 1
**Contribution:** 1
**Rating:** 1
**Confidence:** 4

**Summary:**

In this work, the authors aim to improve network efficiency and reduce parameter redundancy by introducing Replacement Learning, a straightforward approach that fixes the parameters in a layer by interpolating between two adjacent layers. The experimental results indicate that this method is effective—although I find these results somewhat difficult to believe.

**Strengths:**

I’d say the idea is somewhat interesting, and the paper follows the ICLR submission style.

**Weaknesses:**

1. Figure 1 doesn’t make any sense—using very large models on very small datasets. It would be more meaningful to test on a larger dataset like ImageNet, with different models of different scales for comparison.

2. what is the motivation of such a method?

3. There are several unsupported and arbitrary statements in the paper. For example, in Line 83,  "Considering that parameters from adjacent layers, if solely derived from either shallow or deep layers, often fail to simultaneously enable frozen layers to excel in learning both local and global features" lacks justification. The relationship to local or global features is unclear here, as adjacent layers don’t inherently correspond to shallow or deep feature characteristics.

4. In Eq. 9, the parameters of the i-th layer are a linear combination of those from the previous and next layers. What’s the motivation behind? It’s also unclear why this setup would yield better performance, as shown in Table 1 and Table 2.

5. What is the motivation for providing detailed information about backpropagation? I don’t see any differences or novel contributions in this section.

6. In the experiments, I don’t understand how reducing the number of parameters in a model can consistently improve performance across different datasets and models. This seems completely unreasonable and doesn’t make sense at all, especially I didn't see anything can help with this.

7. BTW, the results on CIFAR-10, SVHN, and STL-10 aren’t reliable, as these datasets are too small to provide meaningful insights.

8. Why is k=4 chosen? There should be an ablation study to justify this choice. Additionally, why is there a frozen layer every k layers throughout the networks? I assume we can select layers.

9. What happens if the frozen layer is the last layer in a ResNet stage? How would parameter interpolation be handled in this case?

10. The experimental settings MUST have issues. All the results for the vanilla models are SIGNIFICANTLY lower than original papers. This calls into question the reliability of the results presented in the paper, potentially not only ImageNet.

11. Typo： feature maps can be found in Figure 4.3.1.

12. I cannot tell any essiential differences among the four images in Fig. 3.

**Questions:**

I must express my disappointment after spending several hours reviewing such a submission.

I suggest that the authors reconsider the motivation behind the proposed method and conduct all experiments with greater seriousness and care, as many results currently seem unreasonable.

---

### Official Review · Reviewer_tFZt · 2024-11-04

**Soundness:** 2
**Presentation:** 3
**Contribution:** 2
**Rating:** 3
**Confidence:** 4

**Summary:**

This paper proposes an efficient training method for deep neural networks, named Replacement Learning, which aims to reduce the number of trainable parameters, training time, and memory consumption. Replacement Learning achieves this by selectively freezing the parameters of certain layers, which then utilize parameters from adjacent layers updated through a parameter integration mechanism controlled by just two learnable parameters. This method leverages information from surrounding structures to enhance overall model performance while reducing computation and conserving GPU memory.

**Strengths:**

1. The paper is clearly written and is generally easy to follow.

2. The problem being studied in the paper is becoming increasingly important recently.

3. The idea is simple yet seems to be something that people haven’t tried before.

**Weaknesses:**

1. The proposed method only marginally reduces the GPU memory consumption and training time compared to the baseline training method.

2. The paper did not compare the proposed method with any other parameter-efficient training methods, such as [3].

3. Although the paper discusses related works on alternative backpropagation methods and training utilizing surrounding layers, none of the related works are compared with the proposed methods in the experiments.

4. Parameter-efficient training methods [1, 2] are widely applied in fine-tuning pre-trained networks by selectively updating a small subset of model parameters, streamlining the adaptation process of pre-trained models, and facilitating rapid deployment across various domains. However, this paper only studies the setting for training-from-scratch.

[1] Zhang, Taolin, et al. "Parameter-efficient and memory-efficient tuning for vision transformer: a disentangled approach." ECCV 2024.

[2] He, Xuehai, et al. "Parameter-efficient model adaptation for vision transformers." AAAI 2023.

[3] Mostafa, Hesham, and Xin Wang. "Parameter efficient training of deep convolutional neural networks by dynamic sparse reparameterization." ICML 2019.

**Questions:**

1. How does the value of $k$ impact the performance of the models? The author should perform an ablation study on this value.

2. How does the proposed method compare with other parameter-efficient training methods?

3. How does the proposed method compare with other alternative backpropagation methods and methods utilizing surrounding layers during training?

4. Can this method be applied to fine-tuning pre-trained networks?

---

### Official Review · Reviewer_7Uue · 2024-11-05

**Soundness:** 3
**Presentation:** 3
**Contribution:** 3
**Rating:** 8
**Confidence:** 3

**Summary:**

This paper introduces Replacement Learning, a novel training technique aimed at reducing the number of learnable parameters in deep learning models while maintaining or even enhancing model performance. The approach specifically targets the limitations of traditional end-to-end training, such as high computational demand, memory usage, and parameter redundancy, which are common in deeper architectures. Rather than updating all parameters during backpropagation, Replacement Learning freezes certain layers and uses parameters from adjacent layers, controlled by two learnable parameters, to inform the frozen layers through a parameter integration mechanism. This design enables the frozen layers to leverage both local and global feature representations, balancing historical context with new inputs while reducing memory and computational costs.

The authors conduct experiments across multiple image classification datasets (CIFAR-10, STL-10, SVHN, and ImageNet) using various architectures, including CNNs and Vision Transformers (ViTs). Results demonstrate that Replacement Learning reduces GPU memory usage, training time, and the number of parameters, while achieving higher accuracy than traditional end-to-end training. Furthermore, the method shows versatility, adapting effectively across different architectures and datasets.

**Strengths:**

1. A novel training strategy that replaces frozen layer parameters with a fusion of parameters from neighboring layers, controlled by two learnable parameters, reducing the training load without compromising performance.
2. Replacement Learning shows substantial savings in memory usage and training time and achieves better accuracy than end-to-end training.
3. The method performs well across diverse architectures (CNNs and ViTs) and datasets, suggesting broad applicability.
4. Extensive experiments on benchmark datasets confirm the effectiveness of Replacement Learning in surpassing the performance of standard training approaches.

**Weaknesses:**

I like the paper overall. However, would like to point some weaknesses which the authors have also mentioned in their limitations section:
1. While effective on image-based tasks, the approach has not yet been tested on other domains such as NLP or multimodal tasks, which limits its generalizability.
2. The paper could benefit from a more in-depth discussion of any limitations associated with freezing certain layers and its impact on long-term learning dependencies, especially in deeper networks.

**Questions:**

I like the paper overall and don't have any major questions to ask.

---

### Note · Authors · 2024-11-16

I have read and agree with the venue's withdrawal policy on behalf of myself and my co-authors.